# Prevalence and distribution of *Plasmodium vivax* Duffy Binding Protein gene duplications in Sudan

Safaa Ahmed[1,2], Kareen Pestana[3], Anthony Ford[4], Mohammed Elfaki[1,5], Eiman Gamil[1], Arwa F. Elamin[1], Samuel Omer Hamad[1], Tarig Mohamed Elfaki[1,6], Sumaia Mohamed Ahmed Abukashawa[2], Eugenia Lo[3,7]*, Muzamil M. Abdel Hamid[1]*

1 Institute of Endemic Diseases, University of Khartoum, Khartoum, Sudan, 2 Department of Zoology, Faculty of Science, University of Khartoum, Khartoum, Sudan, 3 Department of Biological Sciences, University of North Carolina at Charlotte, Charlotte, NC, United States of America, 4 Department of Bioinformatics and Genomics, University of North Carolina, Charlotte, NC, United States of America, 5 Department of Microbiology and Parasitology, Faculty of Medicine, Jazan University, Jazan, Saudi Arabia, 6 National Malaria Control Program, Federal Ministry of Health, Khartoum, Sudan, 7 School of Data Science, University of North Carolina at Charlotte, Charlotte, NC, United States of America

* mahdi@iend.org (MMAH); eugenia.lo@uncc.edu (EL)

**Data Availability Statement:** All relevant data are within the manuscript and its Supporting Information files.

## Abstract

*Plasmodium vivax* Duffy Binding Protein (PvDBP) is essential for interacting with Duffy antigen receptor for chemokines (DARC) on the surface of red blood cells to allow invasion. Earlier whole genome sequence analyses provided evidence for the duplications of *PvDBP*. It is unclear whether *PvDBP* duplications play a role in recent increase of *P. vivax* in Sudan and in Duffy-negative individuals. In this study, the prevalence and type of *PvDBP* duplications, and its relationship to demographic and clinical features were investigated. A total of 200 malaria-suspected blood samples were collected from health facilities in Khartoum, River Nile, and Al-Obied. Among them, 145 were confirmed to be *P. vivax*, and 43 (29.7%) had more than one *PvDBP* copies with up to four copies being detected. Both the Malagasy and Cambodian types of *PvDBP* duplication were detected. No significant difference was observed between the two types of duplications between Duffy groups. Parasitemia was significantly higher in samples with the Malagasy-type than those without duplications. No significant difference was observed in *PvDBP* duplication prevalence and copy number among study sites. The functional significance of *PvDBP* duplications, especially those Malagasy-type that associated with higher parasitemia, merit further investigations.

## Introduction

Malaria is a significant public health problem in Sudan and almost 75% of the population is at risk of malaria. *Plasmodium vivax* is one of the prominent malarial parasites in Sudan that can cause severe infection and substantial morbidity. *Plasmodium vivax* and *P. vivax*/*P. falciparum* co-infection are responsible for 11% and 9.1% of total malaria cases, respectively, in Sudan [1–3]. The emergence and marked increase of *P. vivax* poses new challenges to malarial treatment

**Funding:** The author(s) received no specific funding for this work

**Competing interests:** Authors declare that no competing interests exist.

and control in the country. *P. vivax* was previously less common in Sub-Saharan African countries [1]. However, in recent years, the number of *P. vivax* cases has markedly increased across Africa with increasing proportions of severe cases, and infections spreading to new areas where Duffy-negative individuals are predominant [4–10].

The pathology of *P. vivax* infection depends critically on the parasite's ability to recognize and invade human erythrocytes, and multiply leading to clinical signs or symptoms [11]. While *P. falciparum* uses a complex array of receptors to invade human erythrocytes, *P. vivax* merozoites completely depend on the interactions between *P. vivax* Duffy Binding Protein (PvDBP) and the Duffy antigen receptor for chemokines (DARC) expressed on the surface of human erythrocytes [12–15]. Duffy-negative individuals who have low expression of DARC were previously thought to be resistant to *P. vivax* infections [16], but the essentiality of the PvDBP-DARC interaction for *P. vivax* invasion has been recently challenged [17]. Over the past five years, there has been an increasing number of reports on Duffy-negative individuals being infected with *P. vivax* throughout Africa and in South America [6, 18–21]. This changing epidemiology is a serious public health problem as the majority of African populations are Duffy-negatives [21, 22]. The genetic characteristics of *PvDBP* would provide a deeper understanding of the biological mechanisms behind erythrocyte invasion and the functional consequences of *PvDBP* variation [11]. *PvDBP* was previously described as a single copy gene [18], but recent whole genome sequences from field isolates provided evidence for the duplications of this gene [11, 16]. Parasites with two distinct types of *PvDBP* duplications are circulating globally, namely the Malagasy and the Cambodian duplications based on the where they were first described [11]. The high prevalence of *PvDBP* duplications raises the possibility that this structural change could be linked to the ability of *P. vivax* to infect Duffy-negative individuals [6]. Mutations in *PvDBP* in *P. vivax* from Duffy-negative Ethiopians did not lead to binding to Duffy-negative erythrocytes *in vitro* [21]. Other possible alternative mechanisms of erythrocyte invasion are via *PvDBP* gene duplications that allow evasion of host immune responses [23] or via a Duffy-independent pathway that involves the interactions between PvRBP2b and Transferrin receptor 1 [24].

Parasites with multiple *PvDBP* copies have been shown to influence host immune evasion [23]. Increased *PvDBP* copy number may lead to increased mRNA levels and confer protection to *P. vivax in vitro* against invasion inhibition by human monoclonal antibodies targeting region II of PvDBP [23]. However, the extent of invasion inhibition could also be dependent of *PvDBP* sequence polymorphisms and/or Duffy status of host individuals. In Africa, *PvDBP* duplications have been reported in Ethiopia and Madagascar, but such phenomenon is not yet clear in Sudan. Therefore, this study investigated the prevalence and type of *PvDBP* gene duplication, as well as its relationship with demographic and clinical features among *P. vivax* cases in Sudan. Findings would allow comparisons of the distribution and prevalence of *PvDBP* duplications with other African *P. vivax* isolates.

## Methodology

### Ethics statement

Ethical clearance was obtained from Khartoum State Ministry of Health, Sudan (number KMOH-REC-062.2). Verbal informed consent was obtained from each participant and guardians of minor prior to their participation in the study.

### Sample collection and processing

A cross sectional study was conducted between May 2018 to January 2021 in hospitals and health facilities in urban and rural Khartoum including the Gezira Slang and Alsarorab clinics,

River Nile (Northern Sudan) and Al-Obied (Central Sudan). These study sites have similar malaria incidence and low-moderate transmission [25]. A total of 200 whole blood samples from suspected *P. vivax* patients (24 from Khartoum, 53 from Gezira Slang, 44 from Alsarorab, 55 from River Nile, and 24 from Al Obeid) were collected in EDTA tubes. *P. vivax* was diagnosed by microscopic examination of Giemsa-stained thin and thick blood films and/or rapid diagnosis test (SD Bioline, Standard Diagnostics Inc., South Korea). Demographical and clinical data including age, gender, ethnicity, and medical history were recorded with questionnaire. Patients who were infected with other *Plasmodium* species (*P. falciparum*, *P. malariae*, *P. ovale* and/or mixed infection) were excluded from this study. Genomic DNA was extracted from venous blood or dried blood spots (Whatman 3mm filter paper) using ZymoBead Genomic DNA kit (Zymo Research) following the manufacturer's procedures [6]. The concentrations of DNA were determined using a Nanodrop spectrophotometer (UV1visible Nanodrop 1000, Thermo Fisher) prior to PCR. The obtained DNA concentration ranged from 150–700 ng/μl.

## Identification of *P. vivax* by nested and quantitative PCR

Nested PCR was performed in a total volume of 20μl including maxime PCR premix (intron biotechnology, Inc, South Korea), 2.5U i-Taq DNA polymerase, 2.5μM dNTPs, 2μl DNA template, 0.5μl from each primer (10pmol/μl), and 17μl deionized H2O. Species-specific primers that amplify the 18S rRNA amplicon were used to detect *P. vivax* as previously described [26]. The first round of PCR condition was as follow: initial denaturation 94˚C for 2 minutes, followed by 40 cycles of: denaturation 94˚C for 30 seconds, annealing 55˚C for 1 minute, extension 72˚C for 1 minute and final extension 72˚C for 5 minutes. The second round of PCR was as follow: initial denaturation 94˚C for 2 minutes, followed by 40 cycles of: denaturation 94˚C for 30 seconds, annealing 58˚C for 1 minute, extension 72˚C for 1 minute and a final extension 72˚C for 5 minutes. Negative and positive controls were included in each PCR.

In addition to PCR, the SYBR Green qPCR method that amplifies a segment of the 18S rRNA genes of *P. vivax* [27], was used to quantify parasite density of the samples. Amplification was performed in a 20μL reaction containing 10μL of 2x SYBR Green qPCR Master Mix (Thermo Scientific), 0.5 μM of primer (forward: 5'AGAATTTTCTCTTCGGAGTTTATTCT TAGATTGCT-3'; reverse: 5'GCCGCAAGCTCCACGCCTGGTGGTGC-3'), and 1μL of genomic DNA. PCR conditions were as follow: initial denaturation of 95˚C for 3 minutes, followed by 45 cycles of: denaturation at 94˚C for 30 seconds, annealing at 55˚C for 30 seconds, extension at 68˚C for 1 minute immediately followed by 95˚C hold step for 10 seconds and a final melting curve step with temperatures increasing from 65˚C to 95˚C in 0.5˚C increments. Each assay included a positive plasmid control for the 18S rRNA *P. vivax* gene (MRA-178; BEI Resources) to ensure primer specificity in addition to negative controls. A ten-fold dilution series was used to estimate the standard curve and thus amplification efficiency. Patient samples which yielded *Ct* values higher than 40 were considered negative. *P. vivax* parasitemia was calculated with the following equation: Parasite DNA (per/μL) = $[2^{E\times(40-Ctsample)}/10]$; where Ct is the threshold cycle of the individual sample and E is the amplification efficiency [28].

## Duffy blood group genotyping

All *P. vivax* positive samples were included in Duffy blood group genotyping based on Taq-Man qPCR assays. The primers (forward: 5'GGCCTGAGGCTTGTGCAGGCAG-3'; reverse: 5' CATACTCACCCTGTGCAGACAG-3') and dye-labeled probes (FAM: CCTTGGCTCTTA[C] CTTGGAAGCACAGG-BHQ; HEX: CCTTGGCTCTTA[T]CTTGGAAGCACAGG-BHQ) amplified the GATA1 transcription factor-binding site of the *DARC* gene promoter. Amplification

was performed in a 20µL reaction containing 7µL TaqMan Fast Advanced Master mix (Thermo Scientific), 0.5 µM of forward and reverse primers, 0.5 µM of each of the dye-labeled probes, and 1µL of DNA template. PCR conditions were as follows: initial denaturation of 95˚C for 2 minutes, followed by 45 cycles of: denaturation at 95˚C for 3 seconds and annealing at 58˚C for 30 seconds. The fluorescent signals emitted by the dye-labeled probes provided the data for allelic discrimination and determination of the Duffy genotype. The *DARC* gene from a subset of Duffy-positive (C/T and T/T) and all Duffy-negative (C/C) individuals were amplified and sequenced to confirm the genotyping results.

## Detection of *PvDBP* duplications

Five separated amplifications were conducted using published primers designed to detect the different types of *PvDBP* duplications. They included primers as positive controls (BF/BR, AF/AR and AF2/AR2), primers specific for the Malagasy duplication (BF/AR), and primers specific for the Cambodian duplication (BF/AR2) (Table 1).

In addition, *PvDBP* copy number was measured with the SYBR Green qPCR method. The *P. vivax* aldolase gene, which is known to be a single copy gene, was used as an internal reference to calculate the copy number of *PvDBP*. The primers for *PvDBP* duplications amplified between region II and region III of *PvDBP* [6]. Amplification was performed in a 20µL reaction containing 10µL of 2x SYBR Green qPCR Master Mix (Thermo Scientific), 0.5 µM of forward and reverse primers (Table 1), and 1µL of genomic DNA that was standardized to ~50 genomes/µL. PCR conditions were as follow: initial denaturation of 95˚C for 3 minutes, followed by 40 cycles of: denaturation at 95˚C for 30 seconds, annealing at 55˚C for 30 seconds, extension at 68˚C for 1 minute immediately followed by 65˚C hold step for 5 seconds and a final melting curve step with temperatures increasing from 65˚C to 95˚C in 0.5˚C increments. Each assay included a positive plasmid control for *PvDBP* and *Pv adolase* (known to be single copy) along with negative controls. Samples were run in triplicate and the average Ct values for *Pv aldolase* ($Ct_{Pvaldo}$) and *PvDBP* ($Ct_{PvDBP}$) were calculated for each sample. $Ct_{Pvaldo\ cal}$ and $Ct_{PvDBP\ cal}$ were the average difference between $Ct_{Pvaldo}$ and $Ct_{PvDBP}$ obtained from the positive control that contained a single copy of *Pv aldolase* and *PvDBP* gene fragments. Quantification of *PvDBP* duplication was calculated based on the equation from previous studies [6, 29, 30], as follow: $N = 2^{\Delta\Delta Ct \pm SD}$, where $\Delta\Delta Ct = (Ct_{Pvaldolase} - Ct_{PvDBP}) - (Ct_{PvAldolase\ cal} - Ct_{PvDBP\ cal})$. The standard deviation was calculated to be used for the calibrator with the following equation: SD

**Table 1. Information of primers used for PCR and qPCR detection of *PvDBP* duplications based on previous study [11].**

| Primer | Primer sequence (5′→3′) |
| --- | --- |
| BF | TCATCGAGCATGTTCCTTTG |
| BR | TTGCACGTACTCGAAACTCAG |
| AF | CCATAAAAGGTAGGAAATTGGAAA |
| AR | GCATTTTATGAAAACGGTGCT |
| AF2 | ACGCGATGTATCTTCTTTTCA |
| AR2 | TAGAACGCACAGTTATTGGC |
| PvDBP: forward | AGGTGGCTTTTGAGAATGAA |
| PvDBP: reverse | GAATCTCCTGGAACCTTCTC |
| PvAldolase forward | GACAGTGCCACCATCCTTACC |
| PvAldolase reverse | CCTTCTCAACATTCTCCTTCTTTCC |

PCR conditions were as follow: initial denaturation 94˚C for 2 minutes, followed by 35 cycles of: denaturation 94˚C for 20 second, annealing 56˚C for 30 second and extension 72˚C for 1 minutes, followed by a final extension 72˚C for 5 minutes. The PCR condition was based on the published protocol with minor modifications [11].

$= \sqrt{(S^2_{pvdbp}+S^2_{pvaldo}+S^2_{cal})}$. For each sample, the assessment of *PvDBP* copy number was repeated twice for validation.

## Statistical analyses

The normality of data was tested using the Shapiro-Wilk test. For data that departed from normality, nonparametric methods were used for pairwise comparison among groups. All potential significance of differences in parasitemia levels among age groups (under 5 years old, 5–12 years old, and above 12 years old), Duffy groups (C/T, T/T and C/C), as well as infections with and without *PvDBP* duplications were assessed using a one-way non-parametric Kruskal-Wallis test. For each categorical variable (age, Duffy group, and *PvDBP* duplications), we employed a non-parametric multiple comparison procedure using Dunn's test with the Benjamini-Hochberg adjustment to control for false discovery rate. A *p*-value of 0.05 was used to determine significance between groups.

## Results

### Characteristics of the study populations

The age of study participants ranged from 3 months to 70 years old (mean = 21 years old). Males represented 60.2% and females represented 39.8% of the study participants. Patients displayed malaria symptoms including fever, headache, vomiting, fatigue, chills, and abdominal pain. All malaria-infected individuals were microscopic-positive and had parasitemia based on microscopy ranged from 182–38,400 (average 6,863±5,911) parasites per microliter of blood. A significant difference was detected in parasitemia levels across age groups. Individuals above 12 years old showed a significantly lower parasitemia than those under 5 years old (*p* = 0.001) as well as those aged 5–12 (*p* = 0.005; Fig 1A).

Of the 200 samples, 145 were confirmed to be *P. vivax*, of which 134 were monoinfection and 11 were coinfected with *P. falciparum* (S1 File). Based on Duffy genotyping and *DARC* gene sequencing, among these 145 *P. vivax*-infected patients, 127 (87.6%) were heterozygous Duffy-positive (C/T), 15 (10.3%) were homozygous Duffy-positive (T/T), and 3 (2.1%) were homozygous Duffy-negative (C/C). Heterozygous Duffy-positive (C/T) individuals were the

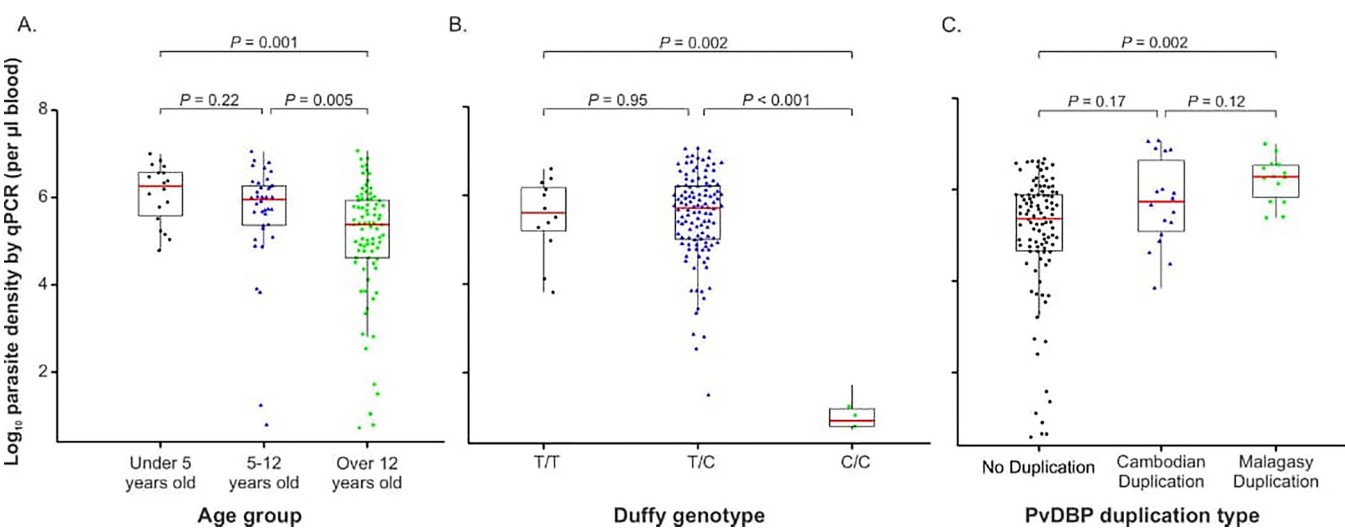

**Fig 1.** Comparisons of parasitemia levels among (A) age groups; (B) Duffy genotypes; and (C) *PvDBP* duplication types based on PCR analysis.

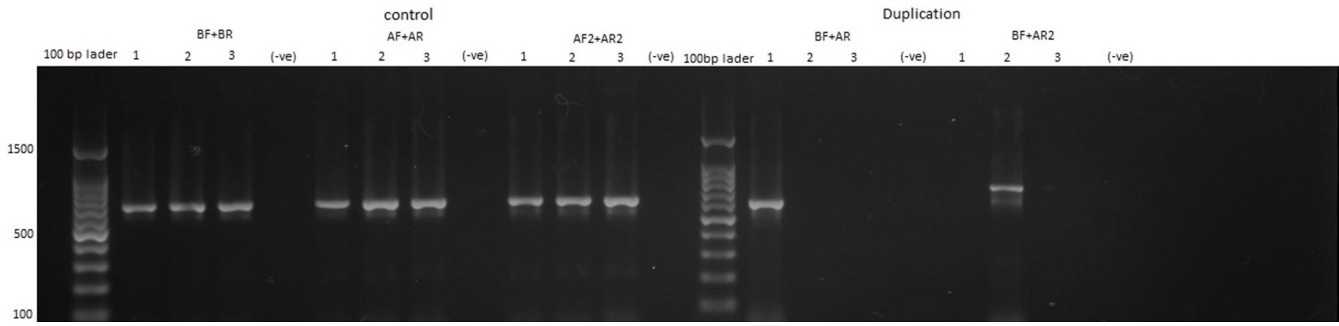

**Fig 2. Gel identification of *PvDBP* duplication types of three selected *P. vivax*-infected samples *based on PCR* assays.** The three positive controls (AF/AF, BF/BR, and AF2/AR2) showed approximately 650bp band, while no bands in the negative controls. Sample (1) indicated Malagasy-type duplication with 613bp band; sample (2) indicated Cambodian-type duplication with 736bp band; and sample (3) indicated a single copy of the *PvDBP* gene with no duplication.

most prevalent. Significant differences were found in parasitemia levels between the Duffy-positive and Duffy-negative groups. The parasitemia was significantly lower in the Duffy-negatives (C/C) than the homozygous Duffy-positive (T/T; $p = 0.002$) and heterozygous Duffy-positive (C/T; $p = 0.001$; Fig 1B), despite small samples of Duffy negatives. No significant difference was detected in parasitemia between the homozygous and heterozygous Duffy-positives ($p > 0.05$). More samples are needed in future study to draw confident comparisons.

### Frequency of PvDBP duplications among Duffy groups and study sites

Of the 145 *P. vivax*-infected samples, 43 (29.7%) were detected with more than one copies of *PvDBP* based on PCR analysis. For these 43 samples with multiple *PvDBP* copies, both the Malagasy and the Cambodian types of *PvDBP* duplication were detected (Fig 2). Samples with the Malagasy-type duplications yielded a 612bp band and those with the Cambodian-type duplications yielded a 736bp band, respectively, using primers BF/AR and BF/AR2 (Fig 2). Three out of the 43 samples with more than one copies of *PvDBP* were observed in *P. vivax* and *P. falciparum* coinfected samples and the rest were *P. vivax* monoinfection.

Of the 43 *P. vivax* samples detected with multiple *PvDBP* copies, 23 (53.5%) have the Cambodian-type and 20 (46.5%) had the Malagasy-type *PvDBP* duplication based on PCR analysis. Approximately 30% of the homozygous and heterozygous Duffy-positives had multiple *PvDBP* copies. The frequency of *PvDBP* duplication was not significantly different between heterozygous (C/T; 36/127 = 28.3%) and homozygous Duffy-positives (T/T; 5/15 = 33.3%) ($p = 0.19$; Table 2). However, only 2 out of the 3 Duffy-negatives (66%) had multiple *PvDBP* copies, and the number of Duffy-negative samples was too small to draw any confident comparisons. For all three Duffy groups, nearly half of the samples with multiple *PvDBP* copies had the Cambodian-type duplications and the other half had the Malagasy-type duplications

**Table 2. Prevalence of *PvDBP* duplications in Duffy-positives (C/T and T/T) and Duffy-negative (C/C) individuals (*N* = 145) *based on PCR analysis*.**

| Duffy genotype | Sample size | Single copy DBP | | Multicopy DBP | |
|---|---|---|---|---|---|
| | | | Total | Cambodian-duplication | Malagasy-duplication |
| T/T | 15 | 10 (66.7%) | 5 (33.3%) | 2 (40%) | 3 (60%) |
| C/T | 127 | 91 (71.7%) | 36 (28.3%) | 20 (55.6%) | 16 (44.4%) |
| C/C | 3 | 1 (33.3%) | 2 (66.7%) | 1 (50%) | 1 (50%) |
| **Total** | **145** | **102 (70.3%)** | **43 (29.7%)** | **23 (53.5%)** | **20 (46.5%)** |

(Table 2). No significant difference was observed between the two types of duplications among Duffy groups ($p = 0.29$; Table 2).

The parasitemia was significantly higher in samples with the Malagasy-type duplication than those with no duplication ($p = 0.002$; Fig 1C). No significant difference was observed in parasitemia between samples with the Cambodian- and Malagasy-type duplications as well as samples with the Cambodian-type and those without duplication (Fig 1C). When stratified by age group, samples with the Malagasy-type duplications have significantly higher parasitemia than the single-copy ones for individuals of age under 5 and over 12 years old (S1 Fig). No significant difference was observed in parasitemia between samples with Cambodian duplication type and those with Malagasy-type or no-duplication. No samples under 5 years old were detected with the Cambodian-type duplications (S1 Fig).

No significant difference was detected in the frequency of *PvDBP* duplications among study sites based on PCR analysis (Table 3). About 36% (13/36) of the *P. vivax* samples in Alsarorab, 33% (6/18) in El Obeid, 30% (11/37) in Gezira Slang, 33% (2/6) in Khartoum, and 23% (11/48) in River Nile were shown with multiple *PvDBP* copies. Based on qPCR analysis, most of the samples with *PvDBP* duplications had 2–3 gene copies (Fig 3). For examples, in Alsarorab, 25% of the samples had 2–3 *PvDBP* copies and 5.6% had 4 copies or higher. In El Obeid, 16.7% and 5.5% had 2–3 copy and 4 or higher copies, respectively. In Gezira Slang, 16.2% and 13.5% had 2–3 copy and 4 or higher copies. In River Nile, 16.7% and 8.35% had 2–3 copy and 4 or higher copies, respectively. By contrast, in Khartoum, all samples with *PvDBP* duplications (33.3%) had 2–3 copies (Fig 3). Both the Malagasy- and the Cambodian-type duplications were detected in each of the study sites (Table 3). The estimations of *PvDBP* duplications by PCR were, for most part, consistent with the estimations by qPCR, with the exception of three

**Table 3. Prevalence of single and multiple *PvDBP* duplications among the five study sites in Sudan *based on PCR analysis*.**

| Study site | Duffy genotype | Sample size | Single copy DBP | Multicopy DBP | Multicopy DBP | |
|---|---|---|---|---|---|---|
| | | | | | Cambodian-duplication | Malagasy-duplication |
| Alsarorab | | | | | | |
| | T/T | 2 | 0 | 2 (100%) | 1(50%) | 1 (50%) |
| | C/T | 33 | 22 (66.7%) | 11 (33.3%) | 4 (36.4%) | 7 (63.6%) |
| | C/C | 1 | 1 (100%) | 0 | - | - |
| El Obeid | | | | | | |
| | T/T | 1 | 1 (100%) | 0 | - | - |
| | C/T | 16 | 11 (86.8%) | 5 (31.2%) | 1 (20%) | 4 (80%) |
| | C/C | 1 | 0 | 1 (100%) | 0 | 1 (100%) |
| Gezira Slang | | | | | | |
| | T/T | 6 | 4 (66.7%) | 2 (33.3%) | 1 (50%) | 1 (50%) |
| | C/T | 30 | 22 (73.3%) | 8 (26.7%) | 7 (87.5%) | 1 (12.5%) |
| | C/C | 1 | 0 | 1 (100%) | 1 (100%) | 0 |
| Khartoum | | | | | | |
| | T/T | 0 | - | - | - | - |
| | C/T | 6 | 4 (66.7%) | 2 (33.3%) | 1 (50%) | 1 (50%) |
| | C/C | 0 | - | - | - | - |
| River Nile | | | | | | |
| | T/T | 6 | 5 (83.3%) | 1 (16.7%) | 0 | 1 (100%) |
| | C/T | 42 | 32 (76.2%) | 10 (23.8%) | 7 (70%) | 3 (30%) |
| | C/C | 0 | - | - | - | - |
| **Total** | | **145** | **102 (70.3%)** | **43 (29.7%)** | **23 (53.5%)** | **20 (46.5%)** |

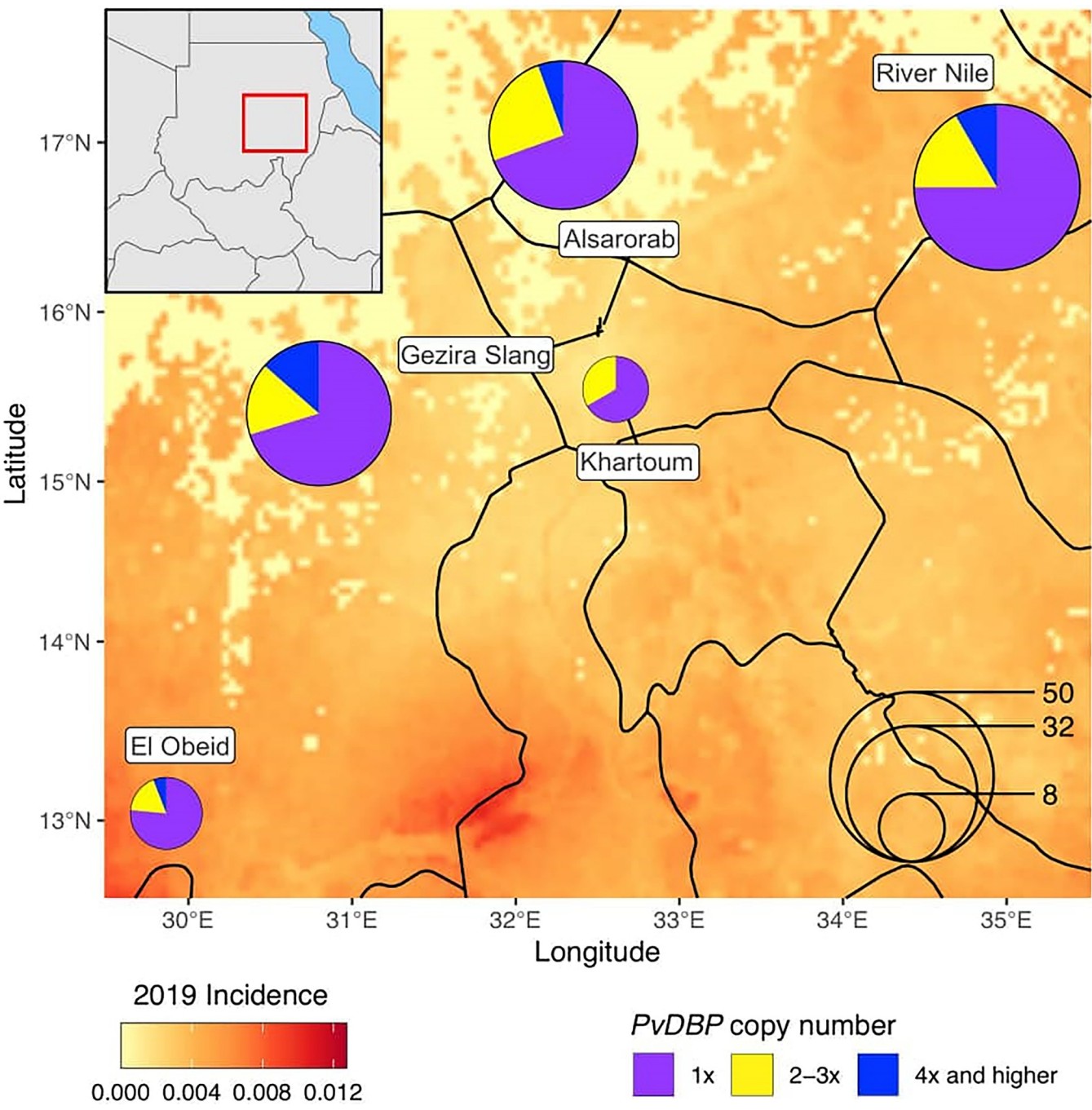

**Fig 3. Distribution of single and multiple *PvDBP* copy parasites among the five study sites in Sudan based on qPCR analysis.**

samples that showed Cambodian/Malagasy-type duplications by PCR but a single-copy by qPCR and four samples that showed high copies by qPCR but no duplication by PCR.

## Discussion

Malarial parasite genomes are highly variable, containing mutations ranging from simple small changes in DNA sequences to complex large-scale changes in gene structure such as the

number of copies of individual genes [11]. These genetic changes allow the parasites to adapt to and/or evade the human immune system and successfully transmit from one to another individual. Information on the structure of genes that involve in red blood cell invasion provides clues for the pathways and mechanisms of parasite invasion. *PvDBP* plays a key role in erythrocyte invasion by *P. vivax* and duplications of this gene might influence the efficiency of the invasion process [11, 31]. This study investigated the prevalence and characteristics of *PvDBP* gene duplications in the Sudanese *P. vivax*.

*PvDBP* duplications occurred in ~30% of the Sudanese *P. vivax*. Both the Malagasy- and the Cambodian-type duplications were detected in the isolates. To our knowledge, this is the first report on the presence of the Cambodian-type *PvDBP* duplication in the Sudanese *P. vivax*. Previous study indicated only the presence of Malagasy-type duplication in Sudan with a relatively low frequency (4/32; 12.5%) [16]. A broader coverage of study areas and larger sample size in this study, as well as the use of specific primer sets that target the Cambodian- and Malagasy-type *PvDBP* duplications may explain such discrepancy. The frequency of the *PvDBP* duplication observed in this study was consistent with a previous study conducted in Cambodia (29%) [11], but lower than that reported in Madagascar (52.9%) [16] and Ethiopia (65.5%) [6]. Given that the same two types of *PvDBP* duplications have been observed among *P. vivax* clinical isolates from Ethiopia and Madagascar [31], it is not surprising that *P. vivax* parasites carrying these duplications have been circulating widely in East Africa [11]. Although the urban and rural areas of Khartoum as well as areas in Northern and Central Sudan are similar in *P. vivax* malaria incidence and transmission intensity [25], the *PvDBP* duplication prevalence was fairly similar among study sites. The discrepancy in *PvDBP* duplications observed in seven samples between PCR and qPCR may be explained by the presence of duplications other than the Cambodian- or Malagasy-type detected by the published primers. Alternatively, mutations at the priming sites may also result in inaccurate assessment of copy number by qPCR. Our ongoing analyses of whole genome sequences from the Sudanese *P. vivax* will provide confirmations to these predictions.

The significantly lower parasitemia observed in individuals over 12 years old compared to other age groups was consistent with earlier studies that younger children are less immune to plasmodial infections [32]. Previous studies showed that the parasite densities were not significantly different between parasites with one or two *PvDBP* copies nor between the two types of *PvDBP* duplications [6, 11]. Nevertheless, we found higher parasitemia in infections with the Malagasy- and Cambodian-type duplications, though the later one is not significantly different from those with no duplication. Such difference prevails when parasitemia was stratified by age group, indicating that age is unlikely a confounding factor for the comparisons. Given only limited samples were detected with Malagasy- or Cambodian-type duplication, future study should expand broader samples to verify these findings. Individuals infected with *P. vivax* parasites carrying multiple copies of *PvDBP* have been shown with high levels of anti-DBPII antibody [23]. *PvDBP* gene amplification leads to increased mRNA levels, and thus it is plausible that high-copies *PvDBP* allows more production of PvDBP proteins that confer to stronger binding with DARC and thus better invasion efficiency than with single-copy parasites [31]. Parasites with high-copies *PvDBP* may also well mediate immune evasion mechanism to enhance erythrocyte invasion [23]. Given only 43 samples were detected with *PvDBP* duplications, the association of parasitemia with duplication types needs to be further verified with larger samples.

Among the *P. vivax* infected samples, heterozygous Duffy-positive (C/T) individuals were the most prevalent, consistent with previous findings [33]. By contrast, the prevalence of Duffy-negative (C/C) individuals (2.1%) was much lower than that reported in Abdelraheem *et al*. (4/48 = 8.3%) [2] and Albsheer *et al*. (34/190 = 17.9%) [33] in Sudan. Individuals of the

three Duffy genotypes were also significantly different in parasitemia, with the highest found in heterozygous Duffy-positives (C/T) and lowest in the Duffy-negatives (C/C). We are currently expanding the number of Duffy-negative samples for copy number assays with the goal to draw confident comparisons with Duffy-positive ones as shown in this study. Low parasitemia in the Duffy-negative individuals has been documented previously [19, 27, 33], and this finding supports the notion that *P. vivax* invasion to human red blood cells is reduced with little to no Duffy antigen expression.

Because of the high adaptability of *Plasmodium* species, *PvDBP* duplications within the parasite genome was likely evolved in response to variations in human Duffy blood group across malaria-endemic settings [31]. The frequency of *PvDBP* duplication was not significantly different between heterozygous Duffy positives (C/T) and homozygous Duffy-positive (T/T), consistent with earlier studies that found no significant difference in the prevalence of *PvDBP* duplications by Duffy genotypes [13, 21, 27]. Previous study on the Ethiopian *P. vivax* showed that the proportion of parasites with *PvDBP* duplications was higher in individuals carrying the FY*A allele than those carrying the FY*B one [27]. *PvDBP* duplications could be a mechanism that increases recognition and/or binding to erythrocytes expressing the FY*A allele through increasing the amount of PvDBP protein on the surface of the merozoites. Though the number of Duffy-negative samples in this study is very small, this result was consistent with an earlier study in Ethiopia where two-third of these infections had *PvDBP* duplications [27]. Arguably, *PvDBP* duplications may not be selected in response to Duffy negativity [29] nor there is a specific sequence polymorphism in *PvDBP* associates with Duffy-negative erythrocyte invasion [34]. *PvDBP* duplications are much more widespread and complex, and the polymorphic nature of *PvDBP* certainly allows *P. vivax* to colonize diverse ecological niches as well as to evade host immune system. Recent studies have shown that *PvDBP* duplications allow *P. vivax* to evade host anti-PvDBP humoral immunity [23], reassuring *PvDBP* region II as a promising candidate for a blood-stage vaccine against *P. vivax*; but whether such structural change influences the PvDBP vaccine efficiency and/or facilitates binding to the weakly expressed DARC on Duffy-negative reticulocytes remain unclear. Further studies are needed to clarify the functional role of *PvDBP* duplications in Duffy-negative erythrocyte invasion.

To conclude, findings of this study allow comparisons of the distribution and prevalence of *PvDBP* duplications with other African *P. vivax* isolates. The origin and functional significance of the Cambodian- and Malagasy-type *PvDBP* duplications merit further investigations. Future study should expand sampling of Duffy-negative infected individuals to test the association between *PvDBP* duplication and Duffy genotypes. It is necessary to determine if *PvDBP* duplication (in relative to single-copy *PvDBP*) is associated with a significant increase in the levels of PvDBP protein expression. More importantly, future studies will be needed to determine if *PvDBP* duplications enable this protein to interact with different invasion receptors on the human RBC surface.

## Supporting information

**S1 File. Information of samples included in this study and *PvDBP* duplication estimations by PCR and qPCR assays.**
(XLSX)

**S2 File. The result of seven *P. vivax*-infected samples which showed discrepancy result between PCR and qPCR.** Gel images of *PvDBP* duplication types of these samples based on PCR were included.
(XLSX)

**S1 Fig. Comparisons of parasitemia levels across *PvDBP* duplication types stratified by age groups based on PCR analysis.**
(PDF)

**S2 Fig. Gel identification of *PvDBP* duplication types of seven *P. vivax*-infected samples that showed discrepancy result between PCR and qPCR. (A)** A 613bp band observed in samples 124, 162, 167 and 168 using the duplication primer BF/AR for Malagasy-type duplication and a 736bp band in samples 18, 55, 99, 133 and 143 using primer BF/AR2 for Cambodian-type duplication. No bands were shown in the negative controls. **(B)** A ~650bp band observed in all samples using the controls primers AF/AR and AF2/AR2. No bands were shown in the negative controls. **(C)** A 650bp band observed in all samples using the controls primer BF/BR. No bands were shown in the negative controls.
(PDF)

**S1 Raw images.**
(PDF)

## Acknowledgments

Authors are grateful for the enrolled participants in all study areas for their participation in this study. M.M.A.H, S.A and E.L kindly provided all materials and consumables for the study.

## Author Contributions

**Conceptualization:** Safaa Ahmed, Eugenia Lo, Muzamil M. Abdel Hamid.

**Data curation:** Safaa Ahmed, Kareen Pestana.

**Formal analysis:** Safaa Ahmed, Kareen Pestana, Anthony Ford.

**Methodology:** Safaa Ahmed, Kareen Pestana, Mohammed Elfaki, Eiman Gamil, Arwa F. Elamin, Samuel Omer Hamad, Tarig Mohamed Elfaki.

**Resources:** Sumaia Mohamed Ahmed Abukashawa, Eugenia Lo, Muzamil M. Abdel Hamid.

**Software:** Anthony Ford.

**Supervision:** Sumaia Mohamed Ahmed Abukashawa, Eugenia Lo, Muzamil M. Abdel Hamid.

**Validation:** Kareen Pestana, Eugenia Lo.

**Writing – original draft:** Safaa Ahmed.

**Writing – review & editing:** Kareen Pestana, Eugenia Lo, Muzamil M. Abdel Hamid.

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
