## [Decision Letter · Decision Letter 0]

7 Dec 2022

PONE-D-22-28855Prevalence and distribution of Plasmodiumvivax Duffy Binding Protein gene duplications in SudanPLOS ONE

Dear Dr. Ahmed,

Thank you for submitting your manuscript to PLoS ONE. After careful consideration, we felt that your manuscript requires substantial revision, following which it can possibly be reconsidered, thus governing the decision of a “major revision”. As requested by the reviewers, the authors need to address several concerns, particularly related to the data analysis, methods and results. For example, considering the discrepancy between methods, it is unclear which results (qPCR vs genotype-specific PCR) were used for data analysis.   In addition, a significant number of issues should be clarified and/or adjust otherwise the MS’s results may be compromised. For your guidance, a copy of the reviewers' comments was included below.

We look forward to receiving your revised manuscript.

Kind regards,

Luzia H Carvalho, Ph.D.

Academic Editor

PLOS ONE

Journal Requirements:

2.Thank you for stating the following in your Competing Interests section:  

"Authors declare that no competing interests exist."

5. We note that Figure 3 in your submission contain map/satellite image which may be copyrighted. All PLOS content is published under the Creative Commons Attribution License (CC BY 4.0), which means that the manuscript, images, and Supporting Information files will be freely available online, and any third party is permitted to access, download, copy, distribute, and use these materials in any way, even commercially, with proper attribution. For these reasons, we cannot publish previously copyrighted maps or satellite images created using proprietary data, such as Google software (Google Maps, Street View, and Earth). For more information, see our copyright guidelines: http://journals.plos.org/plosone/s/licenses-and-copyright.

a. You may seek permission from the original copyright holder of Figure 3 to publish the content specifically under the CC BY 4.0 license.  

Reviewers' comments:

Reviewer's Responses to Questions

**Comments to the Author**

1. Is the manuscript technically sound, and do the data support the conclusions?

Reviewer #1: Yes

Reviewer #2: Partly

2. Has the statistical analysis been performed appropriately and rigorously? 

Reviewer #1: No

Reviewer #2: Yes

3. Have the authors made all data underlying the findings in their manuscript fully available?

Reviewer #1: Yes

Reviewer #2: Yes

4. Is the manuscript presented in an intelligible fashion and written in standard English?

Reviewer #1: Yes

Reviewer #2: Yes

5. Review Comments to the Author

Reviewer #1: The manuscript PONE-D-22-28855 reports on prevalence and distribution of Plasmodium vivax Duffy Binding Protein (PvDBP) duplications among 145 isolates from diverse areas in Sudan. The main findings are (i) identification of both Malagasy and Cambodian types of PvDBP duplications with almost comparable prevalence, (ii) significantly higher parasite density in patients infected with parasites bearing Malagasy type (but not Cambodian type) than those infected with single-copy PvDBP parasites and (iii) significantly higher parasitemia in parasites having Malagasy-type duplication of PvDBP than those without gene amplification. The manuscript is generally well-written and the information may add some knowledge on this vaccine candidate molecule regarding geographic distribution of PvDBP duplication.

Comments/Suggestions

1. The main concern of this manuscript, as also stated by the authors, is the high discrepancy rate of PvDBP copy number determination between quantitative real time PCR and genotype-specific PCR (16 discrepancy samples out of 43 concordant samples). It is unclear which results (qPCR vs genotype-specific PCR) were used for subsequent analysis. With small sample size in this study, such discrepancy and potentially incorrect determination of copy number variation in this locus may preclude meaningful statistical analysis.

2. Please include the range of parasitemia and standard deviation of the average parasitemia (line 216). The mean parasitemia determined by qPCR would be approximately equivalent to 0.001% by microscopy detection which is around the microscopy detection limit. Did most patients have submicroscopic parasitemia? In P. vivax endemic areas outside Africa, most infected individuals have symptomatic malaria. Please describe the clinical status of the patients recruited in this study and parasitemia status of vivax malaria in Africa and outside Africa since parasitemia is an important parameter in this study.

3. There is no sufficient statistical power to draw a conclusion for the Duffy-negatives (n = 3) (lines 228-230).

4. It is confusing that the magnitudes of parasitemia for Malagasy-type and Cambidian-type bearing parasites are not significantly different while only parasites with the former genotype has significantly higher parasitemia than those having single copy of PvDBP. Please explain.

5. In this study, individuals above 12 years old showed a significantly lower parasitemia than those under 5 years old as well as those aged 5-12 (lines 218-220). Please verify that the analysis of parasitemia among patients infected with single-copy, Malagasy-type and Cambodian-type parasites is not affected by the age of the patients.

6. Line 87, increased mRNA levels per se may not always directly indicate protein expression. Please revise.

7. The references need some attention regarding the format and upper-case/lower-case letters.

Reviewer #2: The manuscript by Ahmed et al. presents a more comprehensive data about the prevalence of PvDBP gene duplication and its implication for P. vivax infection in the Sudanese population. The manuscript needs small improvements to clarify some points to the reader.

Major comments:

1. Provide a brief description about malaria incidence in the areas of study, because this information is relevant to understanding of PvDBP duplication prevalence in the different areas.

2. Line 196. There is an error in the formula presented. The delta CT should be calculated by: ΔCT = CT target – CT reference (Livak & Schmittgen 2001, METHODS 25, 402–408). Then, the delta, delta CT is calculated by ΔΔCT = ΔCT test sample – ΔCT calibrator sample. Please make sure that it is a typo and not an error in the formula used to estimate copy number of pvdbp.

3. Line 243. The authors claim that the parasitemia was significantly higher in samples with the Malagasy-type duplication compared to those without duplication. Have you checked if the age (i. e. confounding factor) could explain this difference since parasitemia is significantly different among age groups? This analysis will be important to support the manuscript´s main finding of higher parasitemia in infections with the Malagasy-type duplication.

4. Lines 325-327. This is contrary to the result that has been shown in Lines 247-249 (“The frequency of PvDBP duplication was not significantly different between heterozygous Duffy positives (C/T; 36/127=28.3%) and homozygous Duffy-positive (T/T; 5/15=33.3%) individuals (p-value=0.19; Table 2), …”). Please clarify this part of the Conclusion session.

6. PLOS authors have the option to publish the peer review history of their article (what does this mean?). If published, this will include your full peer review and any attached files.

Reviewer #1: No

Reviewer #2: No

---

## [Author Response · Author response to Decision Letter 0]

15 Feb 2023

February 15, 2023

Dear Editor,

We submit the revised version of the manuscript entitled “Prevalence and distribution of Plasmodium vivax Duffy Binding Protein gene duplications in Sudan”. We are thankful to the reviewers’ constructive comments/suggestions and have taken all comments to improve this manuscript. All gel image data are provided as Supporting Information. We believe that this manuscript is scientifically valid and technically sound. In this revised version, we address the reviewers’ queries point-by-point. All changes can be viewed by track-changes in the uploaded word file. Our responses to the reviewers’ comments are detailed below in this response letter (reviewers’ comments in italic and our responses in bold).

We sincerely look forward to receiving your decision on this revised manuscript.

Yours Sincerely,

Safaa Ahmed

 

Editor: Comments/Suggestions

Our Response: We have revised the manuscript following PLOS ONE requirements.

2.Thank you for stating the following in your Competing Interests section: 

"Authors declare that no competing interests exist."

Our Response: We have inserted the no-conflict statement towards the end of the manuscript and in the online submission form.

Our Response: We have provided the original uncropped and unadjusted gel images as Supporting Information in S1_raw_images.

Our Response: The full ethics statement is now inserted in the Material & Method section.

5. We note that Figure 3 in your submission contain map/satellite image which may be copyrighted. All PLOS content is published under the Creative Commons Attribution License (CC BY 4.0), which means that the manuscript, images, and Supporting Information files will be freely available online, and any third party is permitted to access, download, copy, distribute, and use these materials in any way, even commercially, with proper attribution. For these reasons, we cannot publish previously copyrighted maps or satellite images created using proprietary data, such as Google software (Google Maps, Street View, and Earth). For more information, see our copyright guidelines: http://journals.plos.org/plosone/s/licenses-and-copyright.

Our Response: The License (CC BY 4.0) is provided in online submission. 

Reviewer #1: Comments/Suggestions

1. The main concern of this manuscript, as also stated by the authors, is the high discrepancy rate of PvDBP copy number determination between quantitative real time PCR and genotype-specific PCR (16 discrepancy samples out of 43 concordant samples). It is unclear which results (qPCR vs genotype-specific PCR) were used for subsequent analysis. With small sample size in this study, such discrepancy and potentially incorrect determination of copy number variation in this locus may preclude meaningful statistical analysis.

Our Response: We used genotype-specific PCR for all subsequent analysis as stated in the result, especially for Figure 3 qPCR was used (P11 to 16. L229-309). We have repeated PCR and qPCR for 16 samples twice and confirm that 7 out of 43 samples showed discrepancy results. We provide the genotype-specific PCR gel images and qPCR data in NEW Supplementary File 2. 

2. Please include the range of parasitemia and standard deviation of the average parasitemia (line 216). The mean parasitemia determined by qPCR would be approximately equivalent to 0.001% by microscopy detection which is around the microscopy detection limit. Did most patients have submicroscopic parasitemia? In P. vivax endemic areas outside Africa, most infected individuals have symptomatic malaria. Please describe the clinical status of the patients recruited in this study and parasitemia status of vivax malaria in Africa and outside Africa since parasitemia is an important parameter in this study.

Our Response: Yes, most patients have microscopic parasitemia, and the range, mean and standard deviation of microscopic parasitemia have been added. The clinical symptoms of the patients are also described (P11; L221-224).

3. There is no sufficient statistical power to draw a conclusion for the Duffy-negatives (n = 3) (lines 228-230).

Our Response: Because only 3 Duffy-negatives (C/C) were included in this study, the sample size is too small for any statistical testing. We are currently expanding Duffy-negative samples for DBP copy number assays with the goal to draw confident comparisons with Duffy-positive ones as described in this study (P19 L373-375).

4. It is confusing that the magnitudes of parasitemia for Malagasy-type and Cambidian-type bearing parasites are not significantly different while only parasites with the former genotype has significantly higher parasitemia than those having single copy of PvDBP. Please explain.

Our Response: We stratify the parasitemia of infections with single, Cambodian, and Malagasy copies by age groups. Among all comparisons, only infections with the Malagasy duplications have significantly higher parasitemia than single-copy ones in age under 5 and over 12 years old (Supplementary Figure 1; P14 L284). Limited samples with the Cambodian and Malagasy duplications may explain such differences.

5. In this study, individuals above 12 years old showed a significantly lower parasitemia than those under 5 years old as well as those aged 5-12 (lines 218-220). Please verify that the analysis of parasitemia among patients infected with single-copy, Malagasy-type and Cambodian-type parasites is not affected by the age of the patients.

Our Response: We have addressed this comment by stratifying the parasitemia analyses by age group as mentioned in the above response.

6. Line 87, increased mRNA levels per se may not always directly indicate protein expression. Please revise.

Our Response: We agree with this comment, and thus we stated that ‘Increased PvDBP copy number may lead to increased mRNA levels’. We also add reference #23: Popovici et al. Nat Commun. 2020;11(1):953 to support this statement.

7. The references need some attention regarding the format and upper-case/lower-case letters.

Our Response: All references are revised.

Reviewer #2: 

Major comments:

1. Provide a brief description about malaria incidence in the areas of study, because this information is relevant to understanding of PvDBP duplication prevalence in the different areas.

Our Response: We have included a brief description on malaria incidence in the study areas in the M&M (P5 L106)

2. Line 196. There is an error in the formula presented. The delta CT should be calculated by: ΔCT = CT target – CT reference (Livak & Schmittgen 2001, METHODS 25, 402–408). Then, the delta, delta CT is calculated by ΔΔCT = ΔCT test sample – ΔCT calibrator sample. Please make sure that it is a typo and not an error in the formula used to estimate copy number of pvdbp.

Our Response: We have cited three references for the calculation of copy number (refs# 6,29 and 30 and checked the accuracy of the equations.

3. Line 243. The authors claim that the parasitemia was significantly higher in samples with the Malagasy-type duplication compared to those without duplication. Have you checked if the age (i. e. confounding factor) could explain this difference since parasitemia is significantly different among age groups? This analysis will be important to support the manuscript´s main finding of higher parasitemia in infections with the Malagasy-type duplication.

Our Response: We stratify the parasitemia of infections with single, Cambodian, and Malagasy copies by age groups. Among all comparisons, only infections with the Malagasy duplications have significantly higher parasitemia than single-copy ones in age under 5 and over 12 years old (Supplementary Figure 1; P14 L284). Limited samples with the Cambodian and Malagasy duplications may explain such differences.

4. Lines 325-327. This is contrary to the result that has been shown in Lines 247-249 (“The frequency of PvDBP duplication was not significantly different between heterozygous Duffy positives (C/T; 36/127=28.3%) and homozygous Duffy-positive (T/T; 5/15=33.3%) individuals (p-value=0.19; Table 2), …”). Please clarify this part of the Conclusion session.

Our Response: We have clarified that the frequency of PvDBP duplication was not significantly different between heterozygous Duffy positives (C/T) and homozygous Duffy-positive (T/T) in the Discussion (P19 L381-384).

---

## [Decision Letter · Decision Letter 1]

27 Mar 2023

PONE-D-22-28855R1Prevalence and distribution of Plasmodiumvivax Duffy Binding Protein gene duplications in SudanPLOS ONE

Dear Dr. Ahmed,

Thank you for submitting your manuscript for review to PLoS ONE. After careful consideration, we feel that your manuscript will likely be suitable for publication if the authors revise it to address additional points raised by the reviewer.  

According to reviewers, there are some specific areas where further improvements would be of substantial benefit to the readers, including methods and data analysis. For your guidance, a copy of the reviewers' comments was included below

We look forward to receiving your revised manuscript.

Kind regards,

Luzia H Carvalho, Ph.D.

Academic Editor

PLOS ONE

Journal Requirements:

Reviewers' comments:

Reviewer's Responses to Questions

**Comments to the Author**

1. If the authors have adequately addressed your comments raised in a previous round of review and you feel that this manuscript is now acceptable for publication, you may indicate that here to bypass the “Comments to the Author” section, enter your conflict of interest statement in the “Confidential to Editor” section, and submit your "Accept" recommendation.

Reviewer #2: (No Response)

2. Is the manuscript technically sound, and do the data support the conclusions?

Reviewer #2: Partly

3. Has the statistical analysis been performed appropriately and rigorously? 

Reviewer #2: N/A

4. Have the authors made all data underlying the findings in their manuscript fully available?

Reviewer #2: Yes

5. Is the manuscript presented in an intelligible fashion and written in standard English?

Reviewer #2: No

6. Review Comments to the Author

Reviewer #2: Additional comments

1. The paper requires revision, both in content and language. Results are frequently reported in a confused way. For example, in “Frequency of PvDBP duplications among Duffy groups…” the initial description of PCR results (gels, etc) was presented after showing complete data in Table 2.

2. This reviewer could not find Suppl. Figure 1. Thus, I could not evaluate one of the points raised by this reviewer concerning age as a confounding factor.

3. Indicate the statistical test performed for each analysis.

4. Please, replace “Multi-DBP” with multicopy DBP, “Single-DBP” to single copy DBP to become clear to the readers.

5. Provide the relative frequencies for Table 2 and 3 to make it easier the comparison between groups.

6. Suppl. Figure 2A presents an error in primers nomenclature.

7. PLOS authors have the option to publish the peer review history of their article (what does this mean?). If published, this will include your full peer review and any attached files.

Reviewer #2: No

---

## [Author Response · Author response to Decision Letter 1]

30 May 2023

May 10, 2023

Dear Editor,

We submit the revised version of the manuscript entitled “Prevalence and distribution of Plasmodium vivax Duffy Binding Protein gene duplications in Sudan”. We are thankful to the reviewers’ constructive comments/suggestions and have taken all comments to improve this manuscript. In this revised version, we address the reviewers’ queries point-by-point. All changes can be viewed by track-changes in the uploaded word file. Our responses to the reviewers’ comments are detailed below in this response letter (reviewers’ comments in italic and our responses in bold).

We sincerely look forward to receiving your decision on this revised manuscript.

Yours Sincerely,

Safaa Ahmed

Journal Requirements:

Our Response: 

All references are checked and revised. Reference 32 was mistakenly deleted in the last submitted version due to sentence rearrangement but has now inserted back in this revision.

Reviewer #2: Additional comments

1. The paper requires revision, both in content and language. Results are frequently reported in a confused way. For example, in “Frequency of PvDBP duplications among Duffy groups…” the initial description of PCR results (gels, etc) was presented after showing complete data in Table 2.

Our Response: 

We have carefully checked and modified the content and language throughtout the text in this version. We have made substantial reorganization of text, tables, and figures in the Results based on to the suggestion (P12-13).

2. This reviewer could not find Suppl. Figure 1. Thus, I could not evaluate one of the points raised by this reviewer concerning age as a confounding factor.

Our Response: 

Suppl. Figure 1 was uploaded again in this revision 

3. Indicate the statistical test performed for each analysis.

Our Response: 

We have described in detail the statistical tests used for age, Duffy groups, and PvDBP duplication comparisons (P10 L202-210).

4. Please, replace “Multi-DBP” with multicopy DBP, “Single-DBP” to single copy DBP to become clear to the readers.

Our Response: 

The words are now replaced in Tables 2 and 3.

 5. Provide the relative frequencies for Table 2 and 3 to make it easier the comparison between groups.

Our Response: 

All relative frequencies are now added in Tables 2 and 3.

6. Suppl. Figure 2A presents an error in primers nomenclature.

Our Response: 

Primer nomenclature was corrected in Supplementary Figure 2A and 2B.

---

## [Decision Letter · Decision Letter 2]

12 Jun 2023

Prevalence and distribution of Plasmodium vivax Duffy Binding Protein gene duplications in Sudan

PONE-D-22-28855R2

Dear Dr. Ahmed,

We’re pleased to inform you that your manuscript has been judged scientifically suitable for publication and will be formally accepted for publication once it meets all outstanding technical requirements.

Kind regards,

Luzia H Carvalho, Ph.D.

Academic Editor

PLOS ONE

Additional Editor Comments (optional):

Reviewers' comments:

Reviewer's Responses to Questions

**Comments to the Author**

1. If the authors have adequately addressed your comments raised in a previous round of review and you feel that this manuscript is now acceptable for publication, you may indicate that here to bypass the “Comments to the Author” section, enter your conflict of interest statement in the “Confidential to Editor” section, and submit your "Accept" recommendation.

Reviewer #2: All comments have been addressed

2. Is the manuscript technically sound, and do the data support the conclusions?

Reviewer #2: (No Response)

3. Has the statistical analysis been performed appropriately and rigorously? 

Reviewer #2: (No Response)

4. Have the authors made all data underlying the findings in their manuscript fully available?

Reviewer #2: (No Response)

5. Is the manuscript presented in an intelligible fashion and written in standard English?

Reviewer #2: (No Response)

6. Review Comments to the Author

Reviewer #2: (No Response)

7. PLOS authors have the option to publish the peer review history of their article (what does this mean?). If published, this will include your full peer review and any attached files.

Reviewer #2: No

---

## [Editor Report · Acceptance letter]

11 Jul 2023

PONE-D-22-28855R2 

Prevalence and distribution of *Plasmodium vivax* Duffy Binding Protein gene duplications in Sudan

Dear Dr. Lo:

I'm pleased to inform you that your manuscript has been deemed suitable for publication in PLOS ONE. Congratulations! Your manuscript is now with our production department. 

Kind regards, 

on behalf of

Dr. Luzia H Carvalho 

Academic Editor

PLOS ONE